# Undigested Food and Gut Microbiota May Cooperate in the Pathogenesis of Neuroinflammatory Diseases: A Matter of Barriers and a Proposal on the Origin of Organ Specificity

**DOI:** 10.3390/nu11112714

**Published:** 2019-11-09

**Authors:** Paolo Riccio, Rocco Rossano

**Affiliations:** Department of Sciences, University of Basilicata, 85100 Potenza, Italy; paoloxriccio@gmail.com

**Keywords:** diet, gut microbiota, inflammation, intestinal barrier, blood-brain barrier, Alzheimer’s disease, Parkinson’s disease, multiple sclerosis, autism spectrum disorders, amyotrophic lateral sclerosis

## Abstract

As food is an active subject and may have anti-inflammatory or pro-inflammatory effects, dietary habits may modulate the low-grade neuroinflammation associated with chronic neurodegenerative diseases. Food is living matter different from us, but made of our own nature. Therefore, it is at the same time foreign to us (*non-self*), if not yet digested, and like us (*self*), after its complete digestion. To avoid the efflux of undigested food from the lumen, the intestinal barrier must remain intact. What and how much we eat shape the composition of gut microbiota. Gut dysbiosis, as a consequence of Western diets, leads to intestinal inflammation and a leaky intestinal barrier. The efflux of undigested food, microbes, endotoxins, as well as immune-competent cells and molecules, causes chronic systemic inflammation. Opening of the blood-brain barrier may trigger microglia and astrocytes and set up neuroinflammation. We suggest that what determines the organ specificity of the autoimmune-inflammatory process may depend on food antigens resembling proteins of the organ being attacked. This applies to the brain and neuroinflammatory diseases, as to other organs and other diseases, including cancer. Understanding the cooperation between microbiota and undigested food in inflammatory diseases may clarify organ specificity, allow the setting up of adequate experimental models of disease and develop targeted dietary interventions.

## 1. Chronic Neurodegenerative Diseases are Associated with Low-Grade Chronic Inflammation

Despite having different etiology and different pathogenic mechanisms, chronic neurodegenerative diseases, such as multiple sclerosis (MS), Alzheimer’s disease (AD), Parkinson’s disease (PD), amyotrophic lateral sclerosis (ALS), and autism spectrum disorder (ASD), all have an inflammatory nature in common [1] (Figure 1). 

Fighting the inflammatory processes that underlie these diseases may reduce their progression and their severity. Inflammation is an innate, non-specific defense process [2,3]. It occurs in response to the presence of foreign material (*non-self*), or as a consequence of tissue damage caused by physical, chemical or biological agents, or by abnormalities such as the failure to eliminate waste or digest nutrients. If the cause of inflammation persists, the inflammation also persists, usually with low intensity, and is called low-grade chronic inflammation. As for the chronic neuroinflammatory diseases, in most cases, the neuroinflammatory state does not originate in the central nervous system (CNS), but is thought to come from a chronic systemic inflammation (CSI) [4,5,6]. Recent evidence suggests that CSI may in turn result from a persistent intestinal inflammation spreading through the intestinal barrier, so as to cause a systemic inflammatory and immune response. The persistence of CSI may allow the pro-inflammatory molecules to cross the blood-brain barrier (BBB) and invade the CNS. Once in the brain they can trigger inflammatory processes by activating microglia and astrocytes and pave the way to specific chronic neurodegenerative diseases [6]. At present, it is not known what directs the inflammation towards the CNS and not to other organs. 

Therefore, propagation of the inflammatory state from the intestine to the CNS involves the breakdown of two biological barriers: the intestinal barrier and the blood-brain barrier (BBB). Both are characterized by the existence of intercellular tight junctions, and absence of fenestrations, hence it is not possible to pass anything that does not have a specific transport system or is not lipophilic. To avoid neurological diseases of intestinal origin, the two barriers must remain intact. 

Here we deal with how pro-inflammatory dietary habits (the protracted intake of food that causes low-grade chronic inflammation) may lead to a persistent condition of increased permeability of the intestinal barrier and therefore contribute to the breakdown of the BBB and to neurodegenerative diseases. We suggest that the attack on the BBB might be ascribed to molecular mimicry, with some brain proteins of the undigested food molecules escaping from the intestine and cooperating with gut microbiota in triggering chronic inflammation and an immune response. 

## 2. Food is an Active Subject

To understand whether food has any effect on human health, we should take into account that the intake of food is not just an intake of energy (calories), but rather represents our sharing of energy with the world which is congenial to us. The energy we take in the form of food does not have an exclusively passive role but is active: it is transformed by the host and transforms it. It is indeed well known that food can cause allergies or intolerances in the host that can even be toxic. An example of potential toxicity is the grass pea flour that contains a neurotoxic amino acid, ODAP, (oxalyldiaminopropionic acid, a structural analogue of the neurotransmitter glutamate). In the long run, ODAP causes a neurodegenerative disease called “neurolathyrism” characterized by paralysis of the lower limbs [7,8].

Fundamentally, food can influence our state of health in two ways: (1) by addressing our metabolism, and/or (2) conditioning the composition of our gut microbiota (Figure 2).

The concerted action of our dietary habits on both metabolism and gut microbiota may cause, prevent or diminish chronic inflammation. As we shall see, chronic inflammation is correlated with the various chronic diseases of our time, including the neurodegenerative ones.

## 3. The Effects of Food on Human Metabolism

To “feel” their surrounding environment and adapt themselves to environmental changes, including the nutritional ones, human cells have specific sensors—transcription factors, nuclear receptors and enzymes [9,10]. Transcription factors and nuclear receptors regulate the transcription of genetic information by binding to specific DNA sequences, while enzymes control specific metabolic pathways and, like the sirtuins, may also be active at the nuclear level [11]. 

How can food control metabolism? Some dietary molecules bind to transcription factors, nuclear receptors and enzymes and modulate their activity. The effect on metabolism can be either towards catabolism or towards anabolism. Catabolism has anti-inflammatory effects, anabolism pro-inflammatory effects. Indeed, it is in the anabolic state that the synthesis of pro-inflammatory molecules may occur. Among the active dietary molecules, polyphenols and long chain *n*-3 PUFA (omega-3 polyunsaturated fatty acids), respectively present in vegetables and fish, are anti-inflammatory, up-regulate the catabolic pathways, inhibit anabolism and inflammation, whereas saturated fatty acids and other dietary factors of the animal diet are pro-inflammatory [9,10].

## 4. The Human Gut Microbiota

The human intestine is colonized by a huge number of different microbes: bacteria, viruses and fungi: the gut microbiota. Their number is at least equal to the number of our cells [12] or maybe even larger [13]. Thanks to the recent introduction of culture-independent techniques, such as whole-genome shotgun metagenomics, it is possible to have very reliable data on microbiota composition and to know more about its great biodiversity [14]. We all have our own microbiota, different from all the others like a fingerprint [15,16,17], a diversity that arises from factors such as host genotype, gender, vaginal or C-section birth, age, environmental non-food conditions, stressors, drugs, diseases and above all dietary habits. As the human gut microbiota depends on so many factors, its composition changes continuously every day. 

There is now much interest in gut microbiota. However, as reported by Lita Proctor, a former coordinator of the Human Microbiome Project at the NIH [18], “most of the research so far has placed too much emphasis on cataloguing species names. We’ve been characterizing the human microbiome as if it were a relatively fixed property to be mapped and manipulated—one that it is separated from the rest of the body”. 

In a healthy condition, the different microbial populations live in a harmonic and mutualistic relationship with themselves and with the host. In this condition, called “eubiosis”, the gut ecosystem is well balanced. The eubiotic state is characterized by high microbial diversity, with a marked prevalence of microbes potentially beneficial to us, as they influence practically all functions in the intestine: digestion and energy harvest, mucosal immunity, integrity of the intestinal barrier, protection from pathogens, production of vitamins and other useful metabolites, such as the short chain fatty acids (SCFAs). Moreover, in the eubiotic state the composition of gut microbiota is beneficial for the functionality of all organs of the host, including the brain. 

On the other hand, when the mutual relationship between the different microbial populations is lost, we have a condition called “dysbiosis”. In this condition, the overall biodiversity of the gut microbial system is drastically reduced, and potentially pathogenic microbes prevail over beneficial microbes. The trend to dysbiosis is often indicated by the increase in the Firmicutes (F)/Bacteroidetes (B) ratio, which are the prevalent bacterial phyla in the intestine [19,20].

The eubiotic condition is anti-inflammatory, while the dysbiotic condition is pro-inflammatory. As we shall see, both eubiosis and dysbiosis strongly depend on our life style and in particular on our dietary habits. 

## 5. The Effects of Food on Human Gut Microbiota

The gut microbiota depends on us for its sustenance; indeed, the food we choose to eat determines the composition of the intestinal microbiota, by feeding one or another microbial population, thereby promoting its growth [21,22,23,24].

To explain how food intake determine the composition of our gut microbiota, for the sake of simplicity we can reduce our dietary habits to only two basic types of diet: (1) the vegetarian [VD] diet, which is a low calorie and rich in fiber diet; and (2) the “Western” animal [WD] diet, which is a high calorie diet, mostly carnivorous, rich in saturated fatty acids and refined carbohydrates, and often associated with industrially processed food [25]. The [VD] is suitable only for those gut microbial populations that are able to digest the complex carbohydrates of the fibers, which we cannot degrade. Fibers are the right fuel for bacteria belonging to the phylum Bacteroidetes, such as the Prevotella. In return, they produce useful molecules, such as SCFAs, in particular butyrate [26]. 

[VD] bacteria are associated with intestinal eubiosis, high biodiversity, integrity of the intestinal barrier and human health. The case of the Western diet, which is low in fiber, is of course different. The Western diet is preferred by the microbial populations that do not have the machinery for the digestion of the complex carbohydrates present in fibers. The feeding of the [WD] bacteria, that are more suitable for harvesting energy taken in excess with the Western diet, leads to the production of bile acids, toxic for many [VD] bacteria. Therefore, the mutual relationship between the different microbial populations is lost. The result is a significant decrease in microbiota biodiversity and a dysbiotic state, which is the premise for intestinal inflammation, if it persists over time. To avoid dysbiosis and intestinal inflammation, a predominantly vegetarian diet should be preferred. It is worth pointing out at this point that in healthy conditions, food interacts with gut microbiota in the intestine even if it is not completely digested, while its effects on the metabolism only occur through completely digested molecules, after absorption and therefore outside the intestine.

## 6. Pro-Inflammatory and Anti-Inflammatory Diets

As shown in Figure 3, the energy-dense Western diet is a typically pro-inflammatory diet that is high in saturated animal fats; red meats; fries, snacks and margarines (trans fatty acids); sweetened drinks and simple sugars; salt; processed food; and elaborate condiments. The Western diet is often associated with a sedentary lifestyle and is characterized by the scarcity of fibers. Alcohol intake and smoking are also pro-inflammatory.

Processed foods are pro-inflammatory because they may contain several added chemical additives: artificial flavors, colorants, preservatives, emulsifiers, antibiotics and, in addition, heavy metals, pesticides and herbicides, which have deleterious effects on intestinal microbiota and vitamin D levels.

Figure 4 shows the active factors of the anti-inflammatory diet, mainly vegetarian and rich in fiber. This diet, which is meant to be low in bread and caseinates, is based on the intake of vegetables, fruits, mushrooms, legumes, fish, shellfish, crustaceans, whole wheat pasta, dark chocolate, low-fat yogurt, spices, extra virgin olive oil, coffee and tea.

The calorie-restricted, vegetarian diet should be integrated with fibers and probiotics, controlled or intermittent fasting, and moderate physical activity. An anti-inflammatory diet, tested in a pilot study of patients with relapsing-remitting (RR) or primary-progressive (PP) MS, was effective in reducing inflammation, as evidenced by the decrease in the active forms of MMP-9 (matrix metalloproteinase-9, or gelatinase B) [27].

## 7. What Food Is and Why It must be Digested

Food is what we eat: everything that has to do with the matter of life, not inorganic matter. We do not eat sand, mud, paper or plastic, but everything we recognize to be safe and in its essence similar to us, i.e., that which is made like us and that we know how to “treat”, metabolize and transform, in order to obtain energy or to replace our altered constituents over time. Therefore, our food is made up exclusively of living matter (which has often been inactivated). 

However, when we consume it, food is completely different from us (*non-self*) and we cannot use any of it as it is. Altogether, dietary macromolecules are so different from us that we must provide for their elimination as soon as they occur outside the gastrointestinal system.

Although the biological cells and the macromolecular structures present in our menus (proteins, membranes, polysaccharides) are different from ours, their basic constituents [the bioelements (C,N,O,H), and simple molecules such as fatty acids, monosaccharides, aminoacids] are the same as those we use (Figure 5). Ultimately, living matter is at the same time both foreign to us (*non-self*) and congenial to us (*self*).

As they are different in origin from ours, tissues, cells and proteins from food cannot be used as they are. They must be degraded to simple molecules by the digestive system in the gastro-intestinal tract (the reaction vessel) and then absorbed. This is why food must be digested before being absorbed: it is *non-self* before digestion and becomes *self* when digestion is complete. Only the completely digested molecules are congenial to us, are recognized as *self* and can enter our metabolism after their absorption. In conclusion, the task of digestion is to make food like us, while absorption is required to make simple molecules available to our metabolism. In just over a day (35–40 h) our food becomes part of us (Figure 6).

## 8. Why We Have an Intestinal Barrier

The entire gastro-intestinal tract, from the mouth to the anus, is impervious to what is inside it. The physiological exceptions are the small dietary molecules, obtained by the digestion process of the food taken, if they are hydrophobic or if they have specific transport systems from the lumen to the internal host milieu. 

The structure of the intestinal barrier is complex. We can distinguish a physical, a biochemical and an immunological barrier. 

### 8.1. The Physical Intestinal Barrier

From the lumen to the internal host milieu, the physical barrier consists of one layer (small intestine) or two layers (large intestine) of mucus, a single layer of epithelial cells and the vascular endothelium. The epithelial cells represent the real insurmountable barrier to all those molecules that do not have a transport system and obviously to all microbial cells in the lumen. This is made possible by the fact that intestinal epithelial cells are closely associated with each other through mutual bonding between specific membrane proteins, such as occludin and claudins in the apical part (tight junctions, TJ), and desmosomes and other proteins in the basal part (adherence junctions, AJ), which hold the structure together to form a physical barrier.

The intestinal epithelial barrier is not a static but a dynamic structure [28], as its tight junctions can be opened and closed in response to exogenous and internal stimuli [29]. Important regulators of the integrity of the intestinal barrier are zonulin and related proteins of the zonula occludens family [30]. In the presence of an augmented number of microbes in the small intestine, activation of zonulin opens the tight junctions, to allow flushes of water to wash out microbes and to have the necessary water for the hydrolytic reactions of digestion. 

The epithelial cell layer also contains the so-called goblet cells. Their function is to secrete a glycoprotein, Mucin 2, which forms the “mucus”, the enormous gelatinous layer on the apical surface of the epithelial cells, to limit their direct interaction with microbes. Notably, this protein is also present in the milk fat globule membrane (MFGM): its role in infants is to assist in the development of the digestive system. Nevertheless, the advisability of takeoff consuming cow’s milk (and the MFGM proteins) in adulthood is questionable [31].

### 8.2. The Biochemical and Immunological Intestinal Barriers

The intestinal barrier is not only a tight physical structure, it includes both a biochemical and an immunological defense line. Their function is to block or inactivate the passage of unwanted cells and molecules through the barrier, if it becomes leaky. Indeed, as the barrier is a dynamic structure, its integrity may be occasionally disrupted. Therefore, the intestinal epithelial layer also contains other cells than the epithelial (and the Goblet) ones, capable of producing and secreting protective substances: i.e., the Paneth cells that release antimicrobial peptides, such as the alpha-defensins, lysozyme and REG3 proteins, and numerous T cells, which are among the first immune cells that encounter microbial and dietary antigens.

However, the actual immunological defense line (both innate and adaptive) is found after the mucus layers and the epithelial barrier in the shape of the T and B cells, macrophages and dendritic cells. Activated B cells, present in the gut associated lymphoid tissue (GALT), serve to reinforce the defenses in the mucus layers, as they secrete IgA against potentially pathogenic microbial antigens, but not against mutually beneficial bacteria.

### 8.3. Why the Gut Barrier Is Necessary

It is generally believed that the integrity of the intestinal mucosal barrier and the immunological barrier are needed to avoid the systemic dissemination of bacteria that populate our intestine but, in reality, the main reason for having an intestinal barrier is to prevent dietary macromolecules from getting into the blood stream undigested. As previously mentioned, the digestive tract contains the largest component of the body’s immune system. The immuno-competent cells located at the border of the intestinal barrier are there to check that no stranger passes, by distinguishing what is *non-self* from what is *self*. The distinction between *non-self* and *self* may be based on the estimation of the mimicry of an examined molecule with already known endogenous molecules. The region of the antigen tested, whose probability of mimicry with *self* or *non-self* is, respectively, tolerated or immuno-dominant. However, as microbes and food are both living matter, things are not so simple. 

We have previously shown that similarity profiles (SP)—i.e., the probability of mimicry—between an antigen and human or microbial protein collection are comparable. What really makes the difference is the *non-self*/*self* SP ratio. As this ratio increases, the epitope stimulates T cells with increasing potency [32]. 

In conclusion, the intestinal barrier must be impermeable primarily to dietary molecules that are not completely digested. This is why it exists. Microbial dissemination must be avoided as well, but this is probably not the reason why the intestinal barrier was formed. The breaking of the barrier allows both microbial molecules and cells to escape from the intestine, as well as undigested dietary molecules and immune-competent molecules and cells. All may trigger a systemic inflammatory response. Therefore the breakdown of the intestinal barrier must be absolutely avoided.

## 9. Impact of Dietary Habits on the Integrity of the Intestinal Barrier

### The Regulation of Intestinal Permeability

Despite mucus protection and ‘reinforcement’ by the tight and the adherent junctions, the epithelial cells, which are in the forefront and hence under stress, must be replaced by new cells in less than five days [29]. This means that the tight junctions must be opened and closed again quickly, while the old cells are extruded and the new ones are inserted. The barrier is highly dynamic while remaining a wall against any intrusion of foreign material. It changes quickly to remain itself. However, in the case of inflammatory events the intestinal barrier can become leaky.

The first evidence of a leaky gut barrier was reported a few decades ago in patients with MS [33,34]. Yacyshynet al. [33] demonstrated that 25% of MS patients under study had increased intestinal permeability, while Reichelt and Jensen [34] observed the presence of IgA and IgG antibodies against gliadin and gluten in MS patients. At that time, intestinal functions and the role of gut microbiota were not yet considered, therefore the presence of the antibodies against gliadin and gluten, was attributed to a specific immune response against these proteins in MS.

Nowadays, it is increasingly clear that the intestinal barrier must remain intact to avoid autoimmune diseases [35]. It is therefore important to avoid or limit food and drugs that may loosen the integrity of the gut barrier, including persistent stressful situations, and to prefer dietary factors that may reinforce the integrity of the gut barrier [28]. 

As shown in Figure 7, the following items loosen the integrity of the tight junctions and increase the permeability of the gut barrier: Western diets, saturated fatty acids, gluten, salt, alcohol and chemical additives present in processed food. The “gut-barrier disruptors” act either directly or by modulation of gut microbiota composition. Non-steroidal, anti-inflammatory drugs and stress may also damage the gastro-intestinal tract. Stress causes deterioration of the barrier by activation of the corticotropic-releasing factor (CRF) mast cell axis. 

However, the following have a protective effect on the barrier: calorie restriction or fasting, prebiotics, probiotics, butyrate (SCFAs), vitamins D and A, flavonoids, omega-3 PUFAs, zinc, mucoprotectors (gelatin tannate and tyndallized probiotics).

## 10. Factors Increasing the Permeability of the Gut Barrier

### 10.1. Gluten

Nowadays, it is known that gluten has a direct action on the mucosal barrier of the intestine [36]. It activates the protein zonulin [30], which in turn loosens the tight junctions and makes the intestinal barrier more permeable. It is now clear that the antibody response observed by Reichelt and Jensen [34] was due to the fact that the gliadin proteins passed through the barrier intact or only partially digested. The increased permeability of the intestinal barrier caused by gluten is particularly evident in individuals with non-celiac gluten sensitivity [36].

What is special about gluten is that it is formed in the presence of water from gliadin and glutenin present in wheat, rye and barley, therefore it is present in food such as bread, pizza, cake, pasta and even beer. It is also often added to processed food [37]. Gluten is resistant to digestion. Fragments of not completely digested gluten may be mistaken for a microbial molecule [36], i.e., a protein of adenovirus [38]. For this reason gluten fragments cause the release of zonulin and the opening of the tight junctions. In a similar way, when in the blood stream, gluten opens another barrier equipped with tight junctions: the blood-brain barrier (BBB). As gluten fragments pass through the intestinal wall, they are recognized as a foreign molecule, similar to a viral protein. Following the opening of the barrier, other undigested dietary molecules and microbes also pass through the barrier. All the invaders trigger an immune response. Antibodies against gluten and gliadin can cross react with some brain proteins and can promote neurodegenerative diseases [39,40].

### 10.2. Alcohol

Chronic alcohol intake promotes bacterial overgrowth and gut dysbiosis [41]. It also alters the integrity of the intestinal barrier by decreasing the levels of the anti-microbial molecule REG3, thus favoring microbial access to the gut mucosa [42]. Moreover, alcohol interferes with the metabolism of fatty acids, proteins and carbohydrates, as it converts NAD^+^ into NADH, and is a pro-inflammatory molecule. 

### 10.3. The Impact of Chemicals Present in Processed Foods on the Gut Barrier and Gut Microbiota

Processed food may contain diverse chemicals added to food in order to improve its stability over time and its appeal to the consumer. The additives may be preservatives, artificial flavorings, colorants, emulsifiers, artificial sweeteners and/or antibiotics. All are deleterious for the human gut microbiota [35,43,44,45,46]. For example, dietary emulsifiers decrease the gut microbiota diversity, favor inflammation and reduce the thickness of the mucus layer. Non-nutrient sweeteners (stevia, aspartame and saccharin) have a bacteriostatic effect on gut microbiota. The intake of antibiotics, which may be present in processed food, decreases microbial diversity too, but in addition may cause resistance to antibiotics. An important problem is the addition to processed foods of components from other foods, such as lactose, sugar, whey proteins, gluten, lactose and casein. These added ingredients can represent an overload of particular foods and can create intolerances.

### 10.4. Effects of Gut Dysbiosis on the Permeability of the Intestinal Barrier

What can damage the integrity of the intestinal barrier is first of all gut dysbiosis, often associated with the increase in the Firmicutes/Bacteroidetes ratio and the decrease in overall microbial diversity [19,20]. Firmicutes and Bacteroidetes are the most represented bacterial phyla in the intestine. Persistent dysbiosis leads to an increase in the Th17/Treg ratio and of the lipopolysaccharide LPS, and triggers intestinal inflammation. As a consequence, the tight junctions loosen and the barrier opens. What is in the lumen comes out and enters the blood stream: namely fragments of undigested food; microbes, pro-inflammatory cytokines such as interleukin 6; and endotoxins such as LPS, an endotoxin that is a marker of the translocation of gram-negative bacteria [47,48]. As a result, systemic endotoxemia, chronic systemic inflammation and chronic inflammatory diseases develop. Since intestinal dysbiosis depends primarily on our dietary habits and our life-style, we are the ones who cause intestinal inflammation, the opening of the intestinal barrier, and the metabolic and chronic diseases of our time. Among them it is possible to associate to gut dysbiosis the development of neurodegenerative diseases, which have an inflammatory basis. 

Specific dysbiotic intestinal conditions have been reported in the following neuro inflammatory diseases: Alzheimer’s disease [49,50,51]; Parkinson’s disease [52,53,54]; autism spectrum disorders [20,55]; multiple sclerosis [25,56] and amyotrophic lateral sclerosis (ALS) [57].

## 11. Microbiota and Barrier Protection

### 11.1. Fasting

There have been times when man regularly had no food, whether from spontaneous food gathering, from hunting or agriculture; often it was necessary to fast for several days. It can be said that the most frequent experiences were intermittent fasting and starvation. In contrast, the current most frequent experience in the Western lifestyle is the absence of fasting, at least of a non-scheduled or mandatory fasting. After thousands of years, this is the newest experience in the Western world: the absence of fasting or of lack of food. 

However, between the human gut microbiota and its host, it is the microbiota which is more dependent on fasting. The gut microbiota resides mainly in the large intestine and is intended to eat only the leftovers of what is digested and absorbed by the host in the small intestine. Between the host and the microbiota, it is the latter that is most exposed to the lack of food. 

Indeed, as the uptake of simple carbohydrates, fatty acids and proteins is a process that takes place in the small intestine, in which the microbial presence is usually poor or not elevated, the above dietary nutrients are metabolized primarily by the host and may be insufficient for the colonic microbial population. Above all is the amount of the nitrogen present in the proteins that may be limiting. This is not of secondary importance, as the availability of nitrogen is fundamental for living organisms, and even more for the prokaryotes [58]. Therefore, the amount of nitrogen left by the host may influence the composition of gut microbiota favoring or limiting the growth of microbial populations, depending on their nitrogen dependence. This may be the reason why excess protein in the diet may lead to a high nitrogen availability and to a degraded gut microbial ecosystem [58].

The above is an example of fasting which concerns more the gut microbiota as it is a consequence of the availability or not of a vital element, nitrogen, due to the preferential choice of dietary nutrients by the host. Similar cases occur for simple sugars and fats. 

The case of fasting is different for the host. Fasting for several days, fasting mimicking diet (FMD) [59], intermittent fasting, short term fasting, caloric restriction and time restricted feeding [60] are different fasting or food restriction plans that have been recently proposed to improve health. All improve the integrity of the intestinal barrier, induce a higher microbial diversity, and counteract intestinal inflammation [61,62,63]. In mice with experimental autoimmune encephalitis (EAE), FMD increased the Treg/Th17 ratio and reduced CNS damage [64].

An additional property of fasting is the conversion of white adipose tissue into brown adipose tissue (the “beiging” effect). Moreover, FMD, short term fasting and caloric restriction have been shown to activate intestinal stem cells in mice [62,65].

With regard to the gut microbial population, fasting affects the gut microbiota but does not always show the same effects on microbial diversity: its influence seems to depend on the host and the composition of its gut microbiota, before fasting. In toads, mice, pythons and sea bass, fasting led to an increase of Bacteroidetes and exhibited decreases in the abundances of Coprobacillus and Ruminococcus in response to fasting [66]. The increase of Bacteroidetes (*Bacteroides fragilis*, *Bacteroides thetaiotaomicron*), during fasting may be ascribed to their ability to utilize the mucus glycans as a source of food during fasting [67,68,69], although fasting animals may produce less mucus than fed animals [70]. In mice, FMD was found to promote motor function and reduce the loss of dopaminergic neurons in the substantia nigra by increasing the levels of brain-derived neurotrophic factor (BDNF), which is involved in the survival of dopaminergic neurons. FMD also inhibited neuroinflammation. The above effects have been ascribed to changes in gut microbiota composition and to a higher production of gut microbiota metabolites, such as butyrate.

Digestion is a very demanding activity. In case of sufficient reserves, its suspension with fasting can only bring benefits and can facilitate recovery from digestive stress. Even the time between two meals becomes important, especially the absence of food consumption in the evening and night hours. Fasting for at least 12 hours is important. A meal which does not correspond with circadian clocks may exert adverse effects on human health, while a regular period of at least 12 hours fasting in the evening/night may counteract inflammation, increase stress resistance, and positively modulate gut microbiota population, but above all it allows the intestine to rest [59]. 

### 11.2. Feeding the Human Gut Microbiota

As mentioned above, it is possible for the microbiota to be fasting while the host has enough to eat. Likewise, it is possible to suggest a specific diet for the preferential nutrition of the microbiota at the expense of that of the host. In this case the aim must be to foster an optimal mutually beneficial relationship between the microbiota and the host, as it is in the interest of the host to shape the gut microbiota in such a way as to promote its own wellness.

One important strategy to improve human health may be to nurture certain microbes and not others by choosing those nutrients that can be converted by the microbiota into beneficial metabolites and to avoid those nutrients that are converted by the microbiota into detrimental metabolites [71]. 

In defining a diet that is good for microbiota, we must take into account a basic principle, namely that the microbiota eats the remains of what is not used by the host, and it is not used by the host because it is not available or is too abundant. 

Fiber is the primary food for the gut microbiota. Fibers cannot be digested by the host and therefore arrive almost intact to the microbial populations in the colon. The fermentation of the complex carbohydrates of the fibers produces the short chain fatty acids (SCFAs), including butyric acid, which are very useful for the health of the host [26]. At low concentrations, butyrate improves the epithelial barrier function and counteracts intestinal inflammation by inhibiting the pro-inflammatory transcription factor NF-kB [72] The recommended intake of fiber should be around 25 g/day, and in any case above 15 g/day. 

Other important dietary factors for the microbiota are the phytochemicals. These not only have a very low bioavailability (1–5%) but have no role in our metabolism, except for their anti-inflammatory action, and are seen by our metabolism as foreign molecules. Polyphenols, from vegetables and fruits, and glucosinolates, from vegetables such as cabbage, can be metabolized by gut microbiota and may provide useful molecules such as equol (from daidzein, soya bean) and isothiocyanates, respectively [71]. The intake of polyphenols and their derivatives may correspond to a reduction of the Firmicutes/Bacteroidetes ratio and the preferential growth of beneficial bacteria [73]. 

Other important products obtained in the colon from polyphenols and glucosinolates are the ligands for the aryl hydrocarbon receptors (AHR). AHR are transcription factors integrating environmental, dietary, microbial and metabolic signals with the immune system. Their importance is increasingly being recognized in neurological diseases [74].

On the other hand, shifting from a vegetarian diet to an energy dense Western diet, based on the intake of saturated fat, meat and eggs, and only a little fiber and few polyphenols, may provide substrates to the microbiota, such as carnitine and choline. Trimethylammine (TMA) is a detrimental product of their metabolism in the gut, converted into its oxide TMAO in the liver. TMAO may be associated with cardiovascular events [75], may also be present in human cerebrospinal fluid [76] and is elevated in Alzheimer’s disease [77]. Furthermore, the carnivorous diet provides high amounts of saturated fatty acids and cholesterol that increase the production of bile acids. Their subsequent conversion by the gut microbiota into deoxycholic acid and lithocolic acid, which are toxic for other, mainly beneficial, gut microbes, may lead to a decrease in microbial biodiversity and gut dysbiosis. 

Finally, to get food to the microbiota and ensure a eubiotic condition, it would be good to eat raw food, obviously especially vegetables (e.g., celery, fennel, green salad, cucumber and onion). In fact, if the matrix of the food is not broken through domestic cooking or industrial processing, the host enzymes cannot access the macromolecules inside. The same may happens if food is ingested in large pieces and not in the form of powder or granules.

### 11.3. Vitamins

It has been recently shown that vitamin A improves the integrity of the intestinal barrier, even in the presence of intestinal inflammation and higher LPS level. It seems to counteract the action of LPS and to enhance the expression of tight junction proteins [78].

However, vitamin A is not sufficient. As reported in our previous review [25], vitamin A and vitamin D have synergistic anti-inflammatory effects and should be administered together. This is not surprising as they are both liposoluble and often present together in the same food. Their nuclear receptors cooperate if both vitamins are bound to them. Shared functions of vitamin A and vitamin D include the enhancement of tight junction proteins, the suppression of IFN-γ and IL-17, and the induction of regulatory T cells (Treg) [79]. Finally, vitamins A and D are effective against chronic inflammation and favor the stability of the intestinal barrier. Their action on microbiota is not direct as their nuclear receptors are expressed only by the host, not by the microbiota. Deficiency in vitamin D leads to disruption of the intestinal barrier, gut dysbiosis and intestinal inflammation [80].

### 11.4. Mucoprotectors

Mucosal protectors can restore the normal function of the intestinal barrier by forming a protective film over the epithelium. Among them, gelatin tannate and xyloglucan have been suggested to protect the intestinal mucosa. In addition, probiotics (mainly lactic acid bacteria and bifidobacteria) [81] or tyndallized probiotics [82] may modulate the production of mucin and the expression of epithelial tight junction proteins [83]. Both gelatin tannate and tyndallized probiotics are intended to provide beneficial effects and have been shown to be very effective in the treatment of acute gastroenteritis [83,84,85].

## 12. From Gut Dysbiosis to the Breakdown of the Blood-Brain Barrier and Brain Inflammation 

### Gut Microbiota and Undigested Food Molecules Cooperate in the Assault on the BBB

At first glance, it may seem strange that a dysbiotic condition in the intestine can lead to the damage of the blood-brain barrier (BBB). However, in the course of protracted gut dysbiosis, the normal cross-talk between the gut and the CNS [86,87,88] is somehow disrupted by the molecules that come out of the intestinal lumen into the blood stream and trigger a chronic systemic inflammation. Formation of antibodies against undigested food molecules, which resemble some brain proteins, may address the pro-inflammatory processes to the BBB and cause its breakdown. Indeed, what had been able to make the intestinal barrier more leaky, can also have the same effect on the BBB.

The site of the BBB is the cerebral capillaries. Their endothelial cells are fused to each other by the tight junctions between the proteins claudins, occludin, zone occludens, as they are in the intestinal barrier. Therefore, the passage of molecules and cells between the blood and the brain is absolutely restricted. Under normal conditions, only hydrophobic molecules and molecules that have a specific transport system (i.e., D-glucose, essential amino acids) can pass through the BBB. A major difference between BBB and the intestinal barrier is that the BBB is surrounded by the pseudopods (the membrane projections) of astrocytes. 

The persistence of gut-derived molecules and cells in the proximity of the BBB may cause its breakdown [89]. LPS, the lipopolysaccharide from gram-negative bacteria and LTA, the lipoteichoic acid from gram-positive bacteria, can bind to the Toll-like receptors (TLRs), TLR2 and TLR4 expressed by the brain endothelial cells. The TLRs are involved in the recognition of self and *non-self*-antigens [90]. Their activation initiates an inflammatory and antigen-specific immune responses through the pro-inflammatory transcription factor NFκB and the MAP kinase pathway. 

But this is not enough: other elements escaping from the inflamed gut can impair the integrity of the BBB, in particular the T cells and other components of the immune system such as antibodies and pro-inflammatory cytokines. Altogether, they impair the integrity of the BBB and may trigger autoimmune disorders. The T cells can cross the BBB only if they are activated. In this regard, an important role can be attributed to the gut microbiota, with its gut-associated lymphoid tissues (GALT). GALT activated T cells allow the migration of autoreactive T cells to the brain [89]. Activated T-helper 17 and IL-17A can disrupt the BBB [91]. 

BBB crossover by the foreign molecules and cells mentioned above leads to the activation of microglia. Microglia regulate the pro-inflammatory activity of astrocytes by the release of two proteins—TGF-α (transforming growth factor-α) and VEGF-B (vascular endothelial growth factor-B)—with opposite effects on astrocytes: VEGF-B drives astrocyte pro-inflammatory activation, while TGF-α dampens it. Activated astrocytes destroy neurons and nerve processes and initiate scar formation. Different regions of the brain can be involved in the pathological processes. Enhancing the production of TGF-α and decreasing that of VEGF-B may reverse the inflammatory process and give rise to self-repair. In this respect, the molecules produced by the gut microbiota may be helpful [92]. Indeed, SCFAs [93,94] and tryptophan derivatives [95] from the gut microbiota may modulate microglia and astrocyte activity. In addition to SCFAs, other molecules that may influence TLRs signaling and suppress neuroinflammation may be docoexanoic acid (DHA), a polyunsaturated omega-3 fatty acid, some probiotics, prebiotics such as fructo- and galacto-oligosaccharides (FOS and GOS) [96]. Accordingly, it can be hypothesized that anti-inflammatory diets [27] with dietary supplements [97] may improve inflammatory neurodegenerative diseases. The whole process from gut inflammation to neuroinflammation is schematically represented in Figure 8.

## 13. Conclusions

### Gut Microbiota and Undigested Food Molecules Cooperate in the Set Up of Neuroinflammatory Diseases

Here, we have shown that pro-inflammatory dietary habits may give rise to a sequence of events triggering neuroinflammation and neurodegenerative diseases. The route to disease involves the participation of both the microbiota and undigested food fragments, as well as requiring the disruption of the intestinal barrier and of the blood-brain barrier. The suggested sequence of events is the following: (1) pro-inflammatory diets, protracted over time, change the composition of gut microbiota and induce gut dysbiosis; (2) the enteric immune system is activated and intestinal inflammation rises, T cells are activated, the Th 17/Treg ratio and LPS level increase; (3) the intestinal barrier becomes leaky and the luminal content (microbes, undigested food molecules, endotoxins, T cells and cytokines) comes out, and triggers a chronic systemic inflammation; (4) the immune response against undigested food fragments resembling brain molecules addresses the pro-inflammatory molecules to the BBB and causes its breakdown; (5) the passage through the BBB of activated pro-inflammatory cells and molecules causes the activation of microglial cells and astrocytes and the onset of inflammatory processes in different brain areas. 

The gut microbiota has a key role in this sequence of events and requires special attention. Accordingly, after a long time in which it was completely neglected, many papers on the microbiota have been published in recent years: 12,900 from 2013 to 2017 and a good 4000 in 2017 alone [98]. As the number of papers published on intestinal microbiota in 2018 was 4735, still increasing compared to the previous year, it is possible to state that the role of the microbiota is increasingly recognized as central to human health. However, if we only gave importance to the gut microbiota, we would be wrong. As a matter of fact, the gut microbiota depends on our diet for its sustenance, and what we choose to eat and how much we eat is decisive for eubiotic or dysbiotic gut microbiota [99]. It is food that initiates gut inflammation and probably addresses the inflammatory autoimmune response to the brain. Therefore, we can say that microbiota and food are related to each other, in sickness and in health.

In the present paper we highlight the possible role of undigested food fragments as pro-inflammatory agents and the importance of the integrity of the two barriers, the intestinal and the BBB, for human health. If the intestinal barrier becomes leaky, fragments of undigested food also escape from the luminal space together with bacteria, endotoxins, immunocompetent molecules and cells. All this material, that was supposed to remain segregated in the intestine, is now in circulation. We are usually concerned with the dissemination of bacteria, but the dissemination of undigested food that passes through the intestinal barrier and goes into the bloodstream should not be overlooked.

In the intestine, not completely digested peptides, though still different from us (*non-self*), were on the way to become like us (*self*), thus their probability of molecular mimicry with our peptides may be increased after a partial digestion. In the course of gut dysbiosis and intestinal inflammation, T cells are activated. As T cells are activated, B cells are switched on to produce antibodies. These antibodies against food antigens may recognize self-antigens and trigger an autoimmune response. For example, it has been suggested that antibodies against wheat and milk proteins in blood donors may contribute to neuroimmune activities [39].

Thus, the undigested peptides and their antibodies may reinforce inflammatory and autoimmune activities, and may address them towards one of the different organs, but cooperation with the microbiota is needed. 

The need for cooperation between gut microbiota and antigens in setting up diseases was shown for the first time by Berer et al. (2011) [100]. In their study, it was shown that the myelin oligodendrocyte glycoprotein (MOG), a myelin autoantigen, was able to trigger experimental autoimmune demyelination, but only in the presence of gut microbiota, in germ-free mice it was not possible to activate T and B cells and there was only a poor production of anti-MOG autoantibodies. Revisiting their paper we can try to give a new interpretation that may serve in the present context, which is about the relationship between food and chronic neuroinflammatory diseases. In light of what we have described, partially digested dietary antigens may be taken into account instead of the autoantigen MOG of the Berer study. According with our model, MOG could be now a food molecule, the same MOG or a similar one such as butyrophilin (BTN). BTN is a major protein of the milk fat globules membrane (MFGM) and cow’s milk BTN is very similar to MOG (64% similarity), induces EAE as MOG does and cross-reacts with MOG antibodies [31]. Once the T cells are activated, the availability of dietary antigens ensures the recruitment and activation of antibody-producing B cells and antibodies may attack those structures sharing a similarity with the dietary antigen.

In conclusion, we suggest that what determines the organ specificity of an autoimmune inflammatory process may depend on the food antigens resembling proteins of the organ being attacked. This applies to the brain and neuroinflammatory diseases, as to other organs and other diseases, including cancer. The role of the microbes and their endotoxins in the brain should be evaluated in depth, as the microbes crossing over the BBB, may continue to reside in the brain. Besides sustaining chronic neuroinflammation, they could alter synaptic connections and brain morphology with diverse modalities in different neurological disorders [101]. We probably have to await new technological tools that could prove the presence of a brain microbiome and its role in neurodegenerative diseases [102,103].

Understanding the cooperation between microbiota and undigested food in inflammatory diseases, may clarify organ specificity and the mechanisms leading to neurological disorders, as well as allow the setting up of adequate experimental models of disease and targeted dietary interventions.

## Figures and Tables

**Figure 1 nutrients-11-02714-f001:**
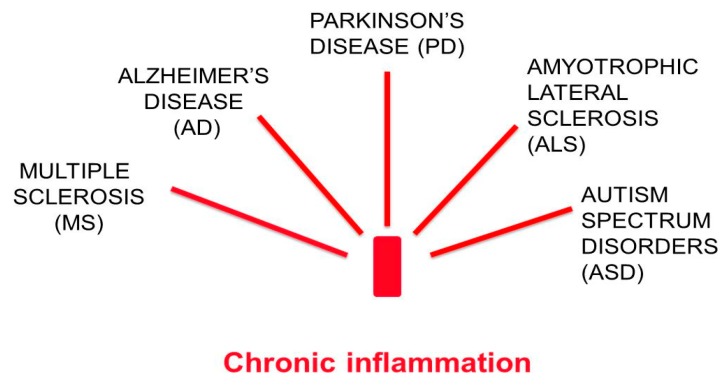
Chronic neurodegenerative diseases have a chronic inflammatory basis in common.

**Figure 2 nutrients-11-02714-f002:**
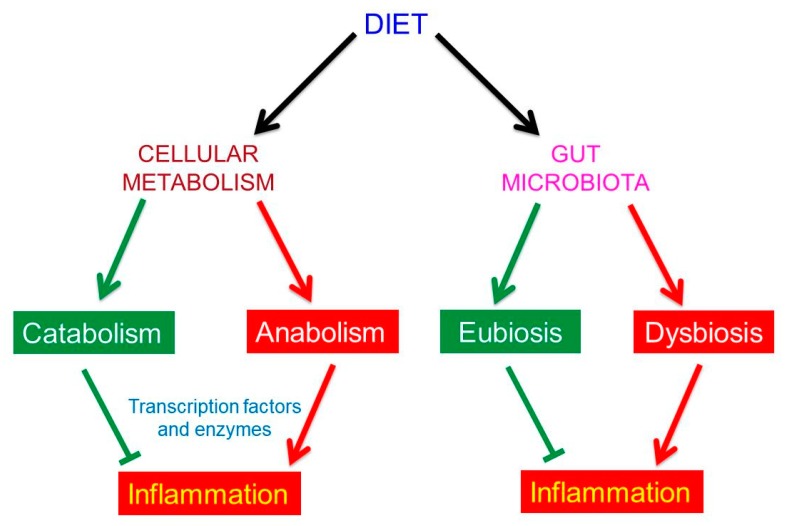
Dietary habits affect both our metabolism and the composition of our gut microbiota.

**Figure 3 nutrients-11-02714-f003:**
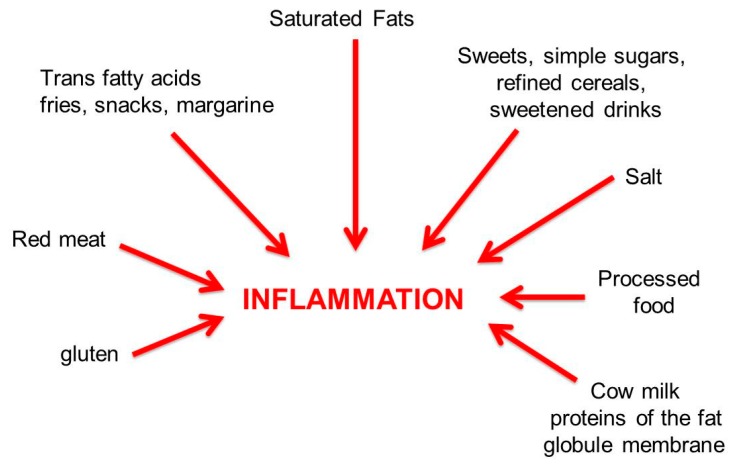
Pro-inflammatory dietary factors.

**Figure 4 nutrients-11-02714-f004:**
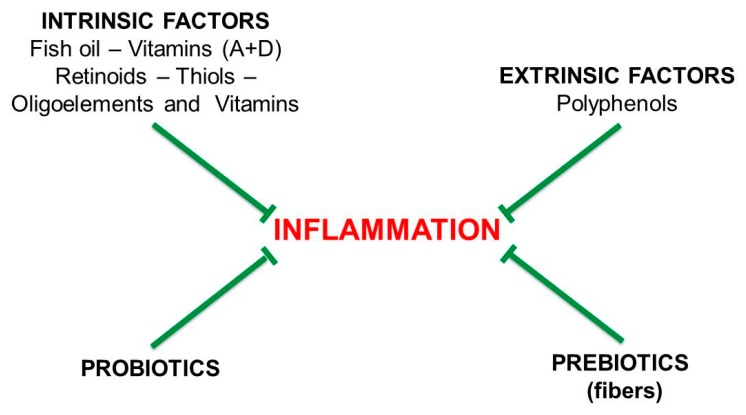
Anti-inflammatory dietary factors. The intrinsic factors are those playing a role in our metabolism. They include: omega-3 poly-unsaturated long-chain fatty acids (*n*-3 PUFAs), present in fish oil; vitamins A and D, B12, PP, E and C; oligoelements such as magnesium, zinc and selenium; thiolic acids such as alfa-lipoic acid (ALA), N-acetyl cysteine and glutathione. The extrinsic factors are the polyphenols, the phytochemicals present in vegetables: they have anti-inflammatory properties and upregulate the catabolism, but are recognized by our metabolism as “foreign” molecules. However, as shown below, they represent a food source for the gut microbiota. Prebiotics and probiotics are cited here for their anti-inflammatory action, but their effects are exerted mainly through the gut microbiota.

**Figure 5 nutrients-11-02714-f005:**
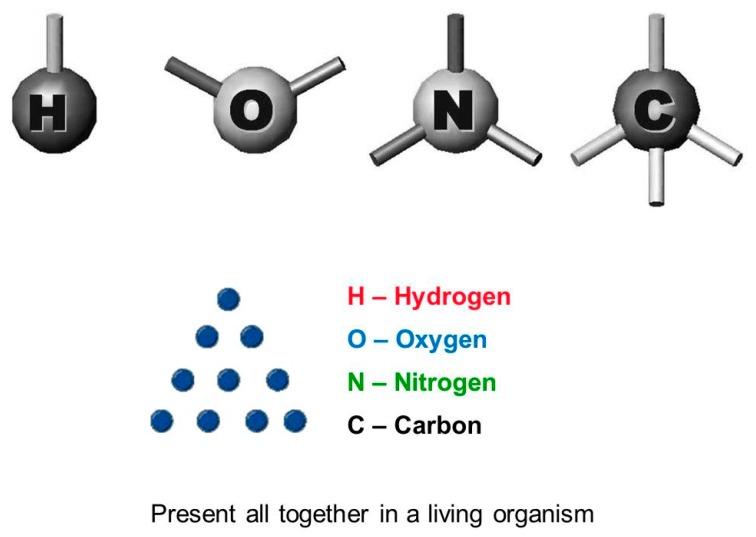
The basic constituents of living matter. The world we know is made up of 92 chemical elements, 81 of which are stable. Living matter uses only about 26–30 of these elements, but 99% of it consists of only four “bioelements”: carbon (C); nitrogen (N); oxygen (O); and hydrogen (H). The bioelements are able to form 4-3-2-1 bonds, respectively, and have a high tendency to get together and form complex molecules such as proteins and nucleic acids, which are different for every species. This means that at the basic level all living organisms are equal to each other, while in their complex forms they are different.

**Figure 6 nutrients-11-02714-f006:**
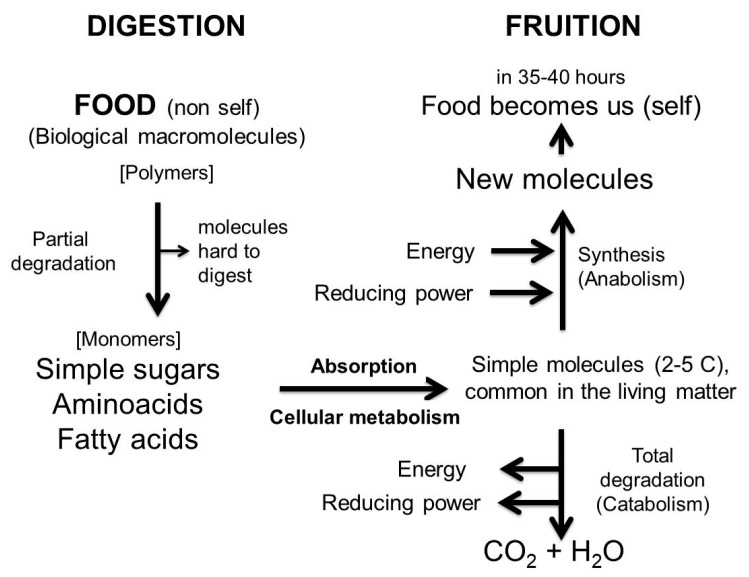
Schematic representation of the metabolic processes, from digestion of the simplest molecules, common to all living organisms and to their fruition.

**Figure 7 nutrients-11-02714-f007:**
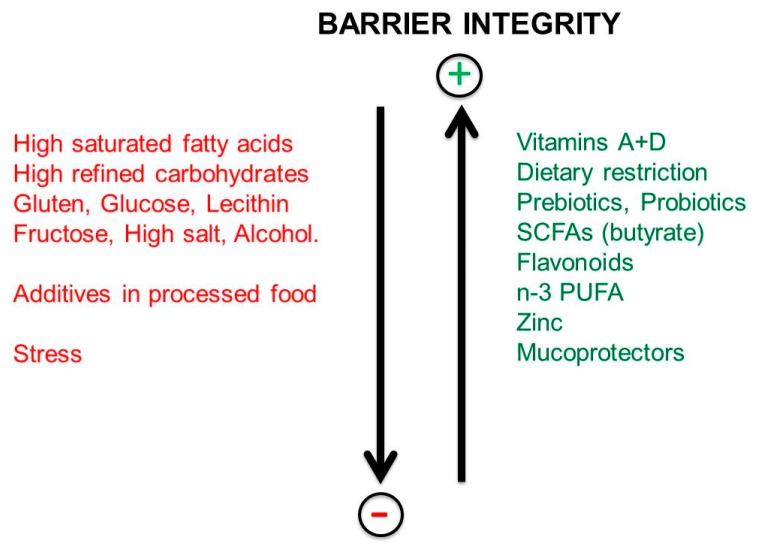
Effects of dietary factors and stressors on the integrity of the intestinal barrier.

**Figure 8 nutrients-11-02714-f008:**
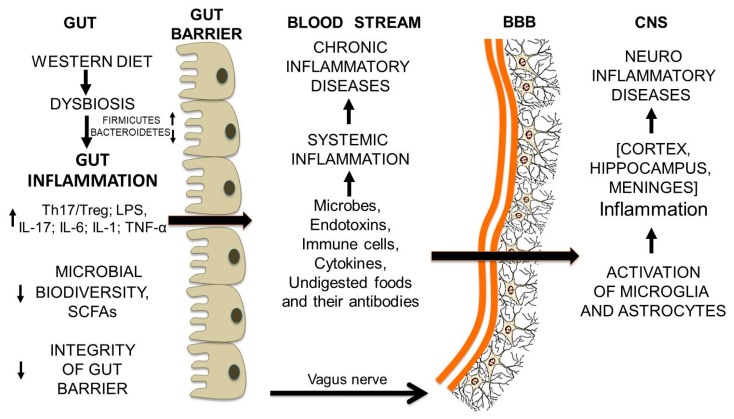
From Westernized dietary habits to neuroinflammation and neurodegenerative diseases: a schematic representation.

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
