# Peer review of "Undigested Food and Gut Microbiota May Cooperate in the Pathogenesis of Neuroinflammatory Diseases: A Matter of Barriers and a Proposal on the Origin of Organ Specificity"

_nutrients, 2019, doi:10.3390/nu11112714_

Round 1
Reviewer 1 Report
Riccio and Rossan’s “Undigested Food and Human Gut Microbiota cooperate in the Pathogenesis of Neuroinflammatory Diseases: A Matter of Barriers” is quite an interesting and well-written review article that includes three different main components. First of all, it is a revision on the established state of the art on the intestinal basis of neuroinflammatory diseases, including a total of 90 references, many of them also reviews themselves. It is highly educational and communicative, with a very plain language. In such a transversal field as gut microbiota, where many specialists need to open their minds to other distant fields, I think this approach is an advantage and fits perfectly in an interdisciplinary journal as Nutrients. Secondly, it can also be considered as a follow up of other previous reviews of theirs (references 4, 5, 7, 19, and 49) spanning from 2011 to 2018 and centred in multiple sclerosis. And thirdly, it is a proposal of the authors on a possible cause of the autoimmune-inflammatory process specificity. It is this latter element that drives my first and main suggestion on the paper. I think it would be interesting and more clarifying if the title included some words in order to imply that it is not only a regular work of reviewing but it includes a hypothesis as well. Maybe something like “Cooperation of undigested Food and Human Gut Microbiota in the Pathogenesis of Neuroinflammatory Diseases: A proposal on a Matter of Barriers”. In this way the reader would know, by just reading the title, that the authors are making a suggestion. Otherwise, it may give the idea that the review is fully talking about proven facts, especially if the reader only accesses the title when referenced in a published paper.
My other main suggestion is on the number of cited works. Certainly, as indicated, there is a sufficient coverage but still there are full paragraphs without any reference, like for instance, the paragraphs included from lines 35 to 59, or when the authors talk about the significance of the Firmicutes/Bacteriodetes ratio without referencing this idea (lines 119 and 344).
Other that that, I have some other minor comments and typos:
- Page 3, line 119: In “Bacteriodetes/(B)”, I think “/” is a typo and it should be “the Firmicutes (F)/Bacteriodetes (B) ratio” (with also a space between (B) and ratio).
- Page 5, line 165: Please, include a space in “factors.Intrinsic”
- Page 5, line 168: “Estrinsic” should be “Extrinsic”.
- Page 6, line 210: “digestion.to” please, remove the dot and include a space.
- Page 7, line 263: “The region of the antigen tested, whose probability of mimicry with self or non-self prevails are, respectively, tolerated or immunodominant.” Please, check this sentence. It seems to me that the subject is singular and should be “is”: “The region of the antigen tested, whose probability of mimicry with self or non-self prevails, is, respectively, tolerated or immunodominant.”
- Page 12, line 507: Please, include the meaning of TGF-α and VEGF-B the first time they are mentioned. It may seem obvious but some readers might not know it.
- Page 13, line 540: Authors mention the number of papers published on intestinal microbiota up to 2017 by citing a 2018 paper of Patrice Cani’s, but since another year has gone by since then, could they include the data for 2018 as well? I just checked it and, according to Pubmed, 4735 papers have been published on intestinal microbiota in 2018, which still indicates an ever-increasing field
Author Response
Reply to Reviewer 1
Comments and Suggestions for Authors
Riccio and Rossano’s “Undigested Food and Human Gut Microbiota cooperate in the Pathogenesis of Neuroinflammatory Diseases: A Matter of Barriers” is quite an interesting and well-written review article that includes three different main components. First of all, it is a revision on the established state of the art on the intestinal basis of neuroinflammatory diseases, including a total of 90 references, many of them also reviews themselves.
Re: We agree with the reviewer and, in order to reduce the number of references to our previous work, we have deleted ref. [7]: Riccio, P.; Rossano, R. The role of nutrition in multiple sclerosis: A story yet to be written. Rev. Esp. Esclerosis Multiple 2013, 5, 24-37 (Page 3).
It is highly educational and communicative, with a very plain language. In such a transversal field as gut microbiota, where many specialists need to open their minds to other distant fields, I think this approach is an advantage and fits perfectly in an interdisciplinary journal as Nutrients. Secondly, it can also be considered as a follow up of other previous reviews of theirs (references 4, 5, 7, 19, and 49) spanning from 2011 to 2018 and centred in multiple sclerosis.
And thirdly, it is a proposal of the authors on a possible cause of the autoimmune-inflammatory process specificity. It is this latter element that drives my first and main suggestion on the paper. I think it would be interesting and more clarifying if the title included some words in order to imply that it is not only a regular work of reviewing but it includes a hypothesis as well. Maybe something like “Cooperation of undigested Food and Human Gut Microbiota in the Pathogenesis of Neuroinflammatory Diseases: A proposal on a Matter of Barriers”. In this way the reader would know, by just reading the title, that the authors are making a suggestion. Otherwise, it may give the idea that the review is fully talking about proven facts, especially if the reader only accesses the title when referenced in a published paper.
Re: We are grateful to the reviewer and, in complete agreement with his suggestion, we have modified the title as follows: “Undigested Food and Human Gut Microbiota may cooperate in the Pathogenesis of Neuroinflammatory Diseases: A Matter of Barriers and a Proposal on the Origin of Organ Specificity”. We have introduced “may” in the title, because the suggested role of undigested food has not proven yet, and have added “and a Proposal on the Origin of Organ Specificity” at the end of the title to highlight our proposal on what can determine organ specificity.
Moreover, in the abstract (line 19) and at the end of the paragraph 1. (line 61) we have added: “We suggest that”. The sentence at the end of paragraph 1 is now: “We suggest that the attack on the BBB might be ascribed to the molecular mimicry with some brain proteins of the undigested food molecules escaped from the intestine and cooperating with gut microbiota in triggering chronic inflammation and immune response.
In the sentence on line 576 we have introduced the adverb "probably" to emphasize that the role of undigested food in determining organ specificity is yet to be demonstrated. The sentence on line 576 is now: “Is food that initiate gut inflammation and probably address the inflammatory autoimmune response to the brain”, instead of: Is food that initiate gut inflammation and address the inflammatory autoimmune response to the brain.”
Rev. 1: My other main suggestion is on the number of cited works. Certainly, as indicated, there is a sufficient coverage but still there are full paragraphs without any reference, like for instance, the paragraphs included from lines 35 to 59, or when the authors talk about the significance of the Firmicutes/Bacteriodetes ratio without referencing this idea (lines 119 and 344).
Re: We strongly agree with the reviewer and have added four new references in the paragraphs from line 35 to 59, and two new references to the Firmicutes/Bacteroidetetes ratio (lines 125 and 368) :
Other that, I have some other minor comments and typos:
Page 3, line 119: In “Bacteriodetes/(B)”, I think “/” is a typo and it should be “the Firmicutes (F)/Bacteriodetes (B) ratio” (with also a space between (B) and ratio). Re: done Page 5, line 165: Please, include a space in “factors.Intrinsic” Re: done Page 5, line 168: “Estrinsic” should be “Extrinsic”. Re: done -Page 6, line 210: “digestion.to” please, remove the dot and include a space. Re: done - Page 7, line 263: “The region of the antigen tested, whose probability of mimicry with self or non-self prevails are, respectively, tolerated or immunodominant.” Please, check this sentence. It seems to me that the subject is singular and should be “is”: “The region of the antigen tested, whose probability of mimicry with self or non-self prevails, is, respectively, tolerated or immunodominant.” Re: Many thanks. Done. Page 12, line 507: Please, include the meaning of TGF-α and VEGF-B the first time they are mentioned. It may seem obvious but some readers might not know it. Re: done Page 13, line 540: Authors mention the number of papers published on intestinal microbiota up to 2017 by citing a 2018 paper of Patrice Cani’s, but since another year has gone by since then, could they include the data for 2018 as well? I just checked it and, according to Pubmed, 4735 papers have been published on intestinal microbiota in 2018, which still indicates an ever-increasing field. Re: done. We agree with the reviewer. Following the citation of Patrice Cani, we have written: “As the number of papers published on intestinal microbiota in 2018 have been 4735, still increasing compared to the previous year, is possible to state that the role of the microbiota is increasingly recognized as central to human health. “ (lines 571-572).

Reviewer 2 Report
Overall, this review could benefit from restructuring, a tighter focus, and better grammar editing.
The subject matter proposed in the abstract is a very exciting area of science and definitely worth writing a review on. There are a lot of exciting concepts brought up, including co-evolution or humans and their microbiome, evolutionary basis for fasting and new intermittent fasting data, and the function of the intestinal barrier. But they could use some further fleshing out; there are gaps in the discussion of these topics, and the flow of the review as a whole feels a bit haphazard and unfocused. A better structure would first discuss microbiome and food interactions in the intestine then strongly connect the gut food processing and microbiome activity to neuroinflammation.
The title of this review says that UNDIGESTED food and the gut microbiota cooperate to achieve neuroinflammation. Much of the review is spent discussing elements of the diet that are pro- or anti-inflammatory, which isn’t quite the same topic – inappropriate exposure to food and/or inappropriate digestion is the most interesting part of this topic, more so than
There is much discussion of how food and certain dietary patterns can be pro- or anti-inflammatory. The focus of this review should stick more specifically to how the microbiome transforms food and/or how disruption of host function by the microbiome contributes to the detrimental impact of undigested food on the brain.
Some of the concepts discussed here are overly simplified. For example, the authors establish a Western diet and calorie-restricted, vegetarian diet as opposite ends of a spectrum. While the need for simplification in this review is apparent, this may perhaps be excessive. For example, studies of calorie-restricted Western diets or low-fiber vegetarian diets have yielded valuable information on the microbiome and inflammation and are relevant for discussion here. Maybe a figure could be helpful with this summary multicomponent plot of different pro- and anti-inflammatory dietary attributes
SPECIFIC AREAS OF DISCUSSION:
Some of the figures are not very useful. An additional figure or two would be useful to better summarize the microbial and food factors travelling through the body to the brain, or regulating immune cell types and other immune phenomena.
Since you're focusing on neuroinflammation, a discussion of recently proposed “brain microbiome” could be useful.
While gluten itself is discussed, this section should be expanded to discuss food intolerances or allergy development based on inappropriate exposure to other partially digested proteins like dairy or egg proteins.
More discussion of how the autoimmune process occurs in the presence of inflammatory agents or undigested molecules could be beneficial.
There is no discussion of differences among hosts playing a role in these processes discussed. Three major gut microbiome enterotypes are known to exist in humans. Humans with different genetic backgrounds or microbiome enterotypes may be better suited to different diets because of how they or their microbiota process food e.g. genetic predisposition to Celiac disease or genotype-based response to dietary saturated fat or salt levels.
The vagus nerve is an interesting area that is not really discussed.
More discussion of similarities between the blood-gut and the blood-brain barrier would also fit into this topic.
The fasting section (11.1) is a very interesting subject but doesn’t seem as relevant to how undigested food and the gut microbiota contribute to neuroinflammation.
GRAMMAR:
There are also quite a few instances of incorrect grammar or tense e.g. “capable to produce” rather than “capable of producing”. It is extremely difficult to discern the meaning of some sentences and paragraphs. There are several instances of run-on sentences with confusing structure and grammar. An example:
“As the intestinal barrier is a dynamic structure and may be more or less intact, even occasionally, the intestinal barrier is not only a physical structure, but also a biochemical and immunological one, necessary to counter what passes in the case of leaky intestine.” and “However, the very effective immune defense is after the epithelial barrier. From the lumen to the internal milieu, after the outer mucus layers, containing antimicrobial proteins, and the epithelial cells sealed by the tight junctions, there are the innate and adaptive immune cells as T and B cells, macrophages and dendritic cells to form an immunological defense barrier.”
Author Response
Reply to reviewer 2
Comments and Suggestions for Authors
Rev. 2: Overall, this review could benefit from restructuring, a tighter focus, and better grammar editing.
The subject matter proposed in the abstract is a very exciting area of science and definitely worth writing a review on. There are a lot of exciting concepts brought up, including co-evolution or humans and their microbiome, evolutionary basis for fasting and new intermittent fasting data, and the function of the intestinal barrier. But they could use some further fleshing out; there are gaps in the discussion of these topics, and the flow of the review as a whole feels a bit haphazard and unfocused.
Rev. 2: A better structure would first discuss microbiome and food interactions in the intestine then strongly connect the gut food processing and microbiome activity to neuroinflammation.
The title of this review says that UNDIGESTED food and the gut microbiota cooperate to achieve neuroinflammation.
Re: When we say undigested we refer to the host (Fig. 6). In fact the food is digested by the host mainly in the small intestine, while the remains are digested by gut microbiota only later in the large intestine.
The interaction between food and gut microbiota is described in the paragraphs 5, 6, and was reported as well in our previous work. The cooperation between undigested food and gut microbiota is described in the paragraph 12.
Rev.2: Much of the review is spent discussing elements of the diet that are pro- or anti-inflammatory, which isn’t quite the same topic – inappropriate exposure to food and/or inappropriate digestion is the most interesting part of this topic, more so than
There is much discussion of how food and certain dietary patterns can be pro- or anti-inflammatory. The focus of this review should stick more specifically to how the microbiome transforms food and/or how disruption of host function by the microbiome contributes to the detrimental impact of undigested food on the brain.
Re: Actually, the sections of the paper dedicated to the pro- or anti-inflammatory properties of dietary factors are only paragraph 6 and figures 3 and 4. This is because we have already published on this topic. It is not so much considering that paragraphs 8 to 11 are on the intestinal barrier.
Regarding how the microbiota transforms food, we have dedicated to this paragraph 11.2 and also the paragraph 5.
Rev. 2: Some of the concepts discussed here are overly simplified. For example, the authors establish a Western diet and calorie-restricted, vegetarian diet as opposite ends of a spectrum. While the need for simplification in this review is apparent, this may perhaps be excessive. For example, studies of calorie-restricted Western diets or low-fiber vegetarian diets have yielded valuable information on the microbiome and inflammation and are relevant for discussion here. Maybe a figure could be helpful with this summary multicomponent plot of different pro- and anti-inflammatory dietary attributes.
Re: We have already used this simplification in our previous work on Neurotherapeutics (ref. 25). Generally the term Western diet is used as an example of a diet that can lead to intestinal dysbiosis and the term vegetarian diet, or even Mediterranean, to indicate a healthier diet than the Western.
SPECIFIC AREAS OF DISCUSSION:
Rev. 2: Some of the figures are not very useful. An additional figure or two would be useful to better summarize the microbial and food factors travelling through the body to the brain, or regulating immune cell types and other immune phenomena.
Re: We have modified Fig. 8, which has also become the graphical abstract, to better show the path from dietary habits to neuroinflammatory diseases.
Rev. 2: Since you're focusing on neuroinflammation, a discussion of recently proposed “brain microbiome” could be useful.
Re: We don't know if we have centered the right topic, since we had no indications on the matter and didn’t find anything in PubMed in the entry “brain microbiome”. However, we have discussed this point in the discussion (lines 615-620), as follows: “The role of the microbes and their endotoxins in the brain should be evaluated in depth, as the microbes overcrossing the BBB may continue to reside in the brain. Besides to sustain chronic neuroinflammation, they could alter synaptic connections and brain morphology with diverse modalities in different neurological disorders [101]. Probably, we have to await that new technological tools could prove the presence of a brain microbiome and its role in neurodegenerative diseases [102,103].”
Rev.2: While gluten itself is discussed, this section should be expanded to discuss food intolerances or allergy development based on inappropriate exposure to other partially digested proteins like dairy or egg proteins.
Re: We agree with the reviewer that would be interesting to discuss about intolerance or allergy. It would be also nice to talk about the exorphins produced by some food as dairy proteins and the gluten fragments, but we tried to focus on the effects of both microbiota and food on the intestinal barrier and then on the blood-brain barrier.
More discussion of how the autoimmune process occurs in the presence of inflammatory agents or undigested molecules could be beneficial.
Re: We made an example in the discussion when mentioning the work of Berer et al. (2011).
There is no discussion of differences among hosts playing a role in these processes discussed. Three major gut microbiome enterotypes are known to exist in humans. Humans with different genetic backgrounds or microbiome enterotypes may be better suited to different diets because of how they or their microbiota process food e.g. genetic predisposition to Celiac disease or genotype-based response to dietary saturated fat or salt levels.
Re: We agree with the reviewer. Each of us may be better suited to different diets or may be more or less prone to some diseases. Certainly everyone has his own microbiota and therefore a different behavior. However, we had to focus on the intestinal barrier and on the BBB, to describe what is food and why it must be digested, and, as well, how a protacted energy dense diet may cause neuroinflammatory diseases. It could even change brain morphology. We would have liked to talk about the effect of food and microbiota on our mood and psycothic symptoms, but it would not have been a right choice.
The vagus nerve is an interesting area that is not really discussed.
Re. Not to mention the vagus nerve was a choice because it is not strictly related to the integrity of the barriers. There will be more need when it comes to diseases like the Parkinson's Disease.
More discussion of similarities between the blood-gut and the blood-brain barrier would also fit into this topic.
Re: Actually, the concept of similarity between the two barriers has been addressed. Their similarity is fundamental for their breakdown and the presence of tight junctions in both barriers has been stressed as necessary.
The fasting section (11.1) is a very interesting subject but doesn’t seem as relevant to how undigested food and the gut microbiota contribute to neuroinflammation.
Re: the paragraph on fasting has been included as it serves to explain the protective effect of fasting on the barrier and to better clarify the relationship between food and gut microbiota.
Rev. 2: GRAMMAR:
There are also quite a few instances of incorrect grammar or tense e.g. “capable to produce” rather than “capable of producing”. It is extremely difficult to discern the meaning of some sentences and paragraphs. There are several instances of run-on sentences with confusing structure and grammar. An example:
“As the intestinal barrier is a dynamic structure and may be more or less intact, even occasionally, the intestinal barrier is not only a physical structure, but also a biochemical and immunological one, necessary to counter what passes in the case of leaky intestine.” and “However, the very effective immune defense is after the epithelial barrier. From the lumen to the internal milieu, after the outer mucus layers, containing antimicrobial proteins, and the epithelial cells sealed by the tight junctions
, there are the innate and adaptive immune cells as T and B cells, macrophages and dendritic cells to form an immunological defense barrier.”
Re: According with the request, we modified both sentences as follows:
Page 7 . Paragraph 8.2.
Previous sentence: “As the intestinal barrier is a dynamic structure and may be more or less intact, even occasionally, the intestinal barrier is not only a physical structure, but also a biochemical and immunological one, necessary to counter what passes in the case of leaky intestine.”
New sentence: “The intestinal barrier is not only a tight physical structure: it includes both a biochemical and an immunological defense line. Their function is to block or inactivate the passage of unwanted cells and molecules through the barrier, if it becomes leaky. This is because the barrier, in addition to being sensitive to inflammation, is a dynamic structure and its integrity may be disrupted, even occasionally”.
Previous sentence: “However, the very effective immune defense is after the epithelial barrier. From the lumen to the internal milieu, after the outer mucus layers, containing antimicrobial proteins, and the epithelial cells sealed by the tight junctions, there are the innate and adaptive immune cells as T and B cells, macrophages and dendritic cells to form an immunological defense barrier”. Activated B cells, present in the gut associated lymphoid tissue (GALT), secrete IgA against potentially pathogenic microbial antigens , but not against mutualistic bacteria. As the defensins, secreted IgA is located in the mucus layer.
New sentence: However, the actual immunological defense line (both innate and adaptive) is found after the mucus layers and the epithelial barrier: T and B cells, macrophages and dendritic cells. Activated B cells, present in the gut associated lymphoid tissue (GALT), serve to reinforce the defenses in the mucus layers, as they secrete IgA against potentially pathogenic microbial antigens, but not against mutualistic bacteria.

Round 2
Reviewer 2 Report
Thank you for addressing my concerns. I recognize your points that in limiting the scope of your article and appreciate your comments about the other topics I mentioned. As I stated before, this is a very exciting topic and it's wonderful to see it being covered!
I appreciate the impact of having multiple small figures throughout the document in order to give context to the surrounding paragraphs. I would still suggest making them a little more graphically friendly, as you have done with Figure 8, perhaps by adding a few colors (red = pro-inflammatory, green = anti-inflammatory, ...).
I still maintain that the autoimmunity concepts discussed in your section 13 (Conclusions) are very exciting and ought to be discussed in more detail than they are earlier on.
Some examples of grammar and spelling issues:
"Have, instead, a protective effect on the barrier: calorie restriction or fasting, prebiotics, probiotics, butyrate (SCFAs), vitamins D and A, flavonoids, omega-3 PUFAs, zinc, mucoprotectors, (gelatin tannate and tyndallized probiotics)."
"How happens this?"
"Noteworthy, it is good to point out that in a healthy condition, food interacts with gut microbiota in the intestine..."
"This diet, which is meant to be short in bread and caseinates..."
"Between the host and the microbiota, it is the latter that is most exposed to the lack of food." Try "susceptible to a" rather than "exposed to the"
"Nitrogen is highly requested by living organisms, especially by prokariotes." 1. "requested" is incorrect. 2. "prokariotes" should be "prokaryotes"
"Different is the case of fasting for the host."
"The one above is an example of fasting..."
"What had been able to make the intestinal barrier more leaky, can do it also with the BBB."
"thryptophan" should be "tryptophan"
"citokines" should be "cytokines" in Fig 8
"and the luminal content (microbes, undigested food molecules, endotoxins, T cells and cytokines) comes out, and triggers a chronic systemic inflammation;"
"The gut microbiota has a key role"
Author Response
Reply to Reviewer n. 2
Rev.: Thank you for addressing my concerns. I recognize your points that in limiting the scope of your article and appreciate your comments about the other topics I mentioned. As I stated before, this is a very exciting topic and it's wonderful to see it being covered!
Rev.: I appreciate the impact of having multiple small figures throughout the document in order to give context to the surrounding paragraphs. I would still suggest making them a little more graphically friendly, as you have done with Figure 8, perhaps by adding a few colors (red = pro-inflammatory, green = anti-inflammatory, ...).
Re: We have added the colors red and/or green to the figures 1-5 and 7, as suggested.
Rev.: I still maintain that the autoimmunity concepts discussed in your section 13 (Conclusions) are very exciting and ought to be discussed in more detail than they are earlier on.
Rev.: Some examples of grammar and spelling issues:
Re: The manuscript was revised by a native English-speaking Lecturer.
"Have, instead, a protective effect on the barrier: calorie restriction or fasting, prebiotics, probiotics, butyrate (SCFAs), vitamins D and A, flavonoids, omega-3 PUFAs, zinc, mucoprotectors, (gelatin tannate and tyndallized probiotics)."
Re: Sentence modified as follows: However, the following have a protective effect on the barrier: calorie restriction or fasting, prebiotics, probiotics, butyrate (SCFAs), vitamins D and A, flavonoids, omega-3 PUFAs, zinc, mucoprotectors (gelatin tannate and tyndallized probiotics).
"How happens this?"
Re: deleted. Sorry for this sentence.
"Noteworthy, it is good to point out that in a healthy condition, food interacts with gut microbiota in the intestine..."
Re: Sentence modified as follows: It is worth pointing out at this point that in healthy conditions, food interacts with gut microbiota in the intestine …..
"This diet, which is meant to be short in bread and caseinates..."
Re: The sentence has been modified in: "This diet, which is meant to be low in bread and caseinates..."
"Between the host and the microbiota, it is the latter that is most exposed to the lack of food." Try "susceptible to a" rather than "exposed to the"
"Nitrogen is highly requested by living organisms, especially by prokariotes." 1. "requested" is incorrect. 2. "prokariotes" should be "prokaryotes"
Re: The sentence has been modified in:
Indeed, as the uptake of simple carbohydrates, fatty acids and proteins is a process that takes place in the small intestine, in which the microbial presence is usually poor or not elevated, the above dietary nutrients are metabolized primarily by the host and may be insufficient for the colonic microbial population. Above all is the amount of the nitrogen present in the proteins that may be limiting. This is not of secondary importance, as the availability of nitrogen is fundamental for living organisms, and even more for the prokaryotes [58]. Therefore, the amount of nitrogen left by the host may influence the composition of gut microbiota favoring or limiting the growth of microbial populations, depending on their nitrogen dependence. This may be the reason why excess protein in the diet may lead to a high nitrogen availability and to a degraded gut microbial ecosystem [58].
"Different is the case of fasting for the host."
Re: Changed in: The case of fasting is different for the host.
"The one above is an example of fasting..."
Re: Changed in: The above is an example of fasting
"What had been able to make the intestinal barrier more leaky, can do it also with the BBB."
Re: Changed in: Indeed, what had been able to make the intestinal barrier more leaky, can also have the same effect on the BBB.
"thryptophan" should be "tryptophan
Re: done
"citokines" should be "cytokines" in Fig 8
Re: done
"and the luminal content (microbes, undigested food molecules, endotoxins, T cells and cytokines) comes out, and triggers a chronic systemic inflammation;"
"The gut microbiota has a key role
The sentences without our reply were not found, probably because modified during English revision.
